# Impact of disorder on the distribution of gate coupling strengths in a spin qubit device

Sathish R. Kuppuswamy[1*], Hugo Kerstens[1], Chun-Xiao Liu[1, 2], Lin Wang[1] and Anton Akhmerov[1]

**1** Kavli Institute of Nanoscience, Delft University of Technology, Delft 2600 GA, The Netherlands
**2** Qutech, Delft University of Technology, Delft 2600 GA, The Netherlands
* rksathish09@gmail.com

August 4, 2022

## Abstract

A scalable spin-based quantum processor requires a suitable semiconductor heterostructure and a gate design, with multiple alternatives being investigated. Characterizing such devices experimentally is a demanding task, with the full development cycle taking at least months. While numerical simulations are more time-efficient, their predictive power is limited due to unavoidable disorder and device-to-device variation. We develop a spin-qubit device simulation for determining the distribution of the coupling strengths between the electrostatic gate potentials and the effective device Hamiltonian in presence of disorder. By comparing our simulation results with the experimental data, we demonstrate that the coupling of the gate voltages to the dot chemical potential and the interdot tunnel coupling match up to disorder-induced variance. To demonstrate the flexibility of our approach, we also analyze an alternative non-planar geometry inspired by FinFET devices.

## 1  Introduction

Spin qubits are an actively developed platform for building a quantum computer with both single- and two-qubit gate fidelity exceeding the error threshold required for achieving fault tolerance [1–3]. In general, this approach follows the Loss and DiVincenzo proposal [4] and traps single electrons in an array of coupled quantum dots. Time-dependent manipulation of chemical potentials and quantum dot couplings using plunger and barrier electrostatic gates implements coherent operations on the electron spins—the quantum gates [5]. Prototype implementations of a few quantum dot devices were demonstrated in several material platforms, for example in GaAs [5–8], Si/SiGe [9–12], Ge/SiGe [13,14] and Si/SiO$_2$ [15–17] heterostructures.

To go beyond a few qubit devices, the ongoing effort is invested into designing and operating larger-scale quantum dot arrays [8,12,18] necessary to implement more complex quantum circuits without compromising qubit performance. For example, the change from GaAs to Si avoids decoherence due to the hyperfine coupling to the nuclear spins [19]. Because the electron effective mass is larger in Si, the electrostatic gates must be smaller and closer to each other due to the higher electron localization [11,12]. Another ongoing development—2D quantum dot arrays [8,14]—improves qubit connectivity, and more directly corresponds to the

error correcting codes. Although such new designs are vital for future progress, they require going beyond the state of the art in device complexity and precision.

Spin qubit gate designs are commonly characterized by performing the full experimental cycle: fabricating, tuning, and then measuring the device. This pipeline can be partially tested using numerical simulations [20]. Numerical simulations begin with discretizing the continuum Hamiltonian which describes the two-dimensional (2DEG) electron gas in a semiconductor heterostructure. The potential landscape of the 2DEG is shaped to host quantum dots by optimizing the electrostatic environment using non-linear optimizers. In this process, electrostatic potential due to gates is calculated by solving Poisson's equation. Finally, the discretized Hamiltonian is diagonalized to obtain eigenenergies and eigenvectors of the electron states confined in quantum dots. Numerical simulations, despite being complex and demanding, require not longer than days to run, and therefore they are still much faster than fabrication and measurement that lasts months.

Structural variation due to the fabrication as well as electrostatic disorder [21,22] limits the usefulness of numerical simulations. In addition, although gates are designed to predominantly control only one parameter, changing the voltage of one plunger or barrier gate affects all the dot Hamiltonian parameters instead of a single chemical potential or a tunnel coupling [23]. Because of these complications, a significant amount of effort is dedicated to automating the tuning of an experimental device using either an effective model combined with a measurement protocol [23] or using a more general machine learning techniques [24–27].

The final tuning step determines the *virtual gates* [23]—linear combinations of physical gates that couple to a single quantum dot Hamiltonian parameter—that enable efficient time-dependent manipulation of spin qubit devices [28]. Here we demonstrate that such gate coupling strengths are efficiently predicted by numerical simulations even in the presence of disorder. We also provide a reference implementation for performing this procedure, available at [29]. Our simulation of an experimental device agrees with the measurements up to the expected uncertainty due to electrostatic disorder. We also demonstrate the flexibility of our approach by applying it to a non-planar geometry inspired by a FinFET transistor setup [17,30]. Furthermore, by implementing relevant optimizations, we can reduce the simulation time of a single device to a few minutes of CPU time.

## 2   Methods

In the continuum approximation, the Hamiltonian of the 2DEG confined in a semiconductor heterostructure is

$$H_{2D} = -\frac{\hbar^2}{2m_e}\left(\partial_x^2 + \partial_y^2\right) + U(x,y), \tag{1}$$

where $m_e$ is the effective mass and $U(x,y)$ is the gate-defined electrostatic potential. We ignore the spin degree of freedom in this work and only focus on finding the relation between gate potential and electrostatic potential of the 2DEG in the presence of disorder. We further discretize Eq. (1) on a two-dimensional regular grid with nearest-neighbor hoppings to form the tight-binding Hamiltonian $H$.

Electrostatic potential $U(\mathbf{r})$ is a solution to the Poisson's equation

$$\nabla \cdot [\epsilon_r(\mathbf{r})\nabla U(\mathbf{r})] = -\frac{\rho(\mathbf{r})}{\epsilon_0}, \tag{2}$$

where $\epsilon_0$ is the vacuum permittivity, and $\epsilon_r$ is the relative permittivity. We apply Dirichlet boundary conditions with potential $V_i$ at the $i$-th gate electrode boundary and $V = 0$ at the simulation boundary. The source term $\rho(\mathbf{r})$ is the charge density which only includes the possible impurity charges

$$\rho(\mathbf{r}) = \sum_{\mathbf{r}_c} e\delta(\mathbf{r} - \mathbf{r}_c), \tag{3}$$

where the summation is carried out over all the positions $\mathbf{r}_c$ of the charged impurities, and $e$ is the elementary charge. Because we are interested in the single particle Hamiltonian, we neglect the interaction of electrons in different quantum dots, which can be added at later steps. This makes the Poisson equation linear. To reduce the finite-size effects, we simulate a sufficiently large volume surrounding the sample. We use the electrostatic solver of Ref. [31] to construct the finite volume mesh of the simulation area and solve the discretized Poisson equation. To improve performance, we use a coarser grid in the empty space. The general solution of Eq. (2) is

$$U(\mathbf{r}) = U_0(\mathbf{r}) + \sum_k V_k U_k(\mathbf{r}), \tag{4}$$

where $U_0(r)$ is the solution with all $V_k = 0$, and $U_0(r) + U_k(r)$ is the solution with $V_l = \delta_{kl}$ and $\delta$ the Kronecker symbol. Recasting the solution in this form allows us to compute the LU-decomposition of the Laplace operator once, and apply it only once per gate electrode and impurity charge distribution.

We transform the lowest few eigenstates of H that correspond to the ground state of each dot into a maximally localized form along x with the projected position operator

$$\hat{\mathbf{P}}_x = \langle \psi_i | \hat{\mathbf{X}} | \psi_j \rangle, \tag{5}$$

where $1 \leq i, j \leq N$, with $N$ the number of quantum dots. The matrix $W$ diagonalizing $\hat{\mathbf{P}}_x$ is the transformation to the basis of the Wannier orbitals maximally localized in the $x$-direction, with the eigenvalues $x_i$ of $\hat{\mathbf{P}}_x$ being the Wannier centers—the expected $x$-positions of these orbitals. We compute the effective Hamiltonian for the quantum dots in the basis of Wannier functions using

$$H_{\text{dots}} = W^\dagger D W, \tag{6}$$

where $D$ is a diagonal matrix with $N$ lowest eigenvalues of H. Reformulating the Hamiltonian $H$ into the sum of a constant term $H_0$ and the term $H_{V_k}$ which changes as a function of the gate k

$$H = H_0 + \sum_k H_{V_k} \Delta V_K, \tag{7}$$

allows us to compute the onsite and off-diagonal elements of $H_{\text{dots}}$ as a function of the gate voltages

$$H_{\text{dots}} = \sum_i \left( \mu_i + \sum_k \frac{\delta \mu_i}{\delta V_k} \Delta V_k \right) |i\rangle \langle i| \tag{8}$$

$$+ \sum_{i,j} \left( t_{ij} + \sum_k \frac{\delta t_{ij}}{\delta V_k} \Delta V_k \right) |i\rangle \langle j|,$$

where $\mu_i$ is the chemical potential of dot $i$ and $t_{ij}$ is the tunnel coupling between the nearest-neighboring dots $i$ and $j$. The partial derivatives $\delta \mu_i / \delta V_k$ and $\delta t_{ij} / \delta V_k$ are the coupling

strengths between the $k$-th gate and the $i$-th chemical potential and the $ij$-th interdot tunnel coupling respectively.

To tune the device into the optimal operating region, we numerically optimize the gate voltages $\boldsymbol{V}$ to trap a single electron state in each quantum dot and to set all interdot couplings to the same value. We achieve this result using the cost function

$$f(\boldsymbol{V}) = \sum_i \left[ \frac{1}{t_0}(\mu_i(\boldsymbol{V}) - \mu_0) \right]^2 + \log^2\left[ \frac{t_{i,i+1}(\boldsymbol{V})}{t_0} \right], \tag{9}$$

where $\mu_0$ is the desired chemical potential and $t_0$ is the desired interdot tunnel coupling. Since $t$ varies exponentially as a function of the barrier height [28], we apply the natural logarithm to the tunnel couplings in Eq. (9), to make the relative significance of two contributions to the cost function comparable.

In presence of charge impurities in concentrations expected in realistic devices, the optimization using the cost function of Eq. (9) fails to tune the device into the proper tunneling regime or even form quantum dots due to the highly nonlinear nature of the problem. To enable convergence in the presence of disorder we add the disorder potential $H_{\mathrm{dis}}$ with a Lagrange multiplier $H_{\mathrm{dis}}\alpha$ to Eq. (7) and add $(\alpha - 1)^2$ to the cost function. In addition, we optimize dots to be confined to the specific regions by using soft thresholds

$$S(x, \delta x) = \begin{cases} (|x| - \delta x)^2 & \text{for } |x| > \delta x, \\ 0 & \text{otherwise,} \end{cases} \tag{10}$$

where $\delta x$ is the desired range. Combining all terms, we end up with the regularized cost function

$$f'(\boldsymbol{V}) \quad = \quad f(\boldsymbol{V}) \;+\; (\alpha - 1)^2 \;+\; \sum_i \left[ S(x_i(\boldsymbol{V}) - x_{i0}, \delta x) + S(y_i(\boldsymbol{V}) - y_0, \delta y) \right], \quad (11)$$

with $y_i$ the $y$-coordinate of the $i$-th Wannier orbital, $(x_{i0}, y_0)$ is the desired position of the $i$-th dot center, and $(\delta x, \delta y)$ the dot position tolerance. Optimizing $f'(\boldsymbol{V})$ has a high success rate and converges close to the tuned point in the majority of cases. We follow the optimization of $f'$ with an additional optimization of $f$ to ensure that the final configuration corresponds to $\alpha = 1$.

## 3   Results

### 3.1   Planar geometry

We benchmark the numerical simulations by comparing them with the measured data of silicon-doped GaAs/AlGaAs heterostructure measured in Ref. [28]. This device consists of a 2DEG with the dopant density $1.9 \times 10^{11} \, \mathrm{cm}^{-2}$. We approximate the electron density of the undepleted 2DEG—the region away from the dots and the gates—to be equal to the dopant density. Plunger and barrier gates are deposited at $90 \, \mathrm{nm}$ above the 2DEG to selectively deplete the 2DEG and form quantum dots. Plunger gates mainly couple to $\mu_i$, while barrier gates control $t_{ij}$. Following Ref. [28] we consider four plunger gates (P1 to P4) and five barrier gates (B1 to B5) to form an array of four quantum dots. We numerically optimize the

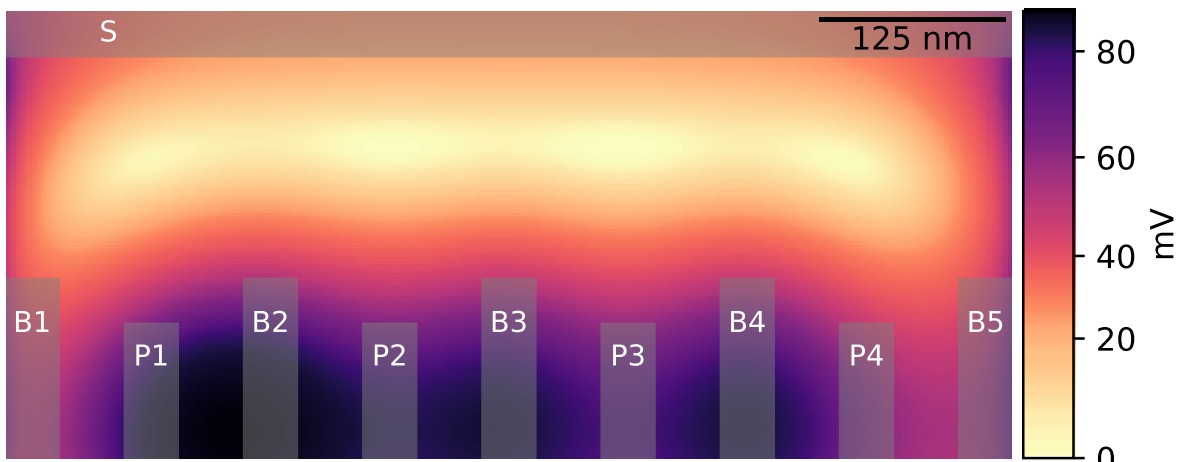

Figure 1: Confinement potential of four quantum dots in the plane of 2DEG. Plunger gates (P1, P2, P3, and P4) are used to tune the chemical potential of each dot, and barrier gates (B2, B3, and B4) are used to tune the interdot tunnel couplings. The screening gate (S) forms a tunnel barrier between the qubit dots and sensor dots (not shown in the image). Gates B1 and B5 control the electron tunneling rate between a corner dot and a neighboring electron reservoir.

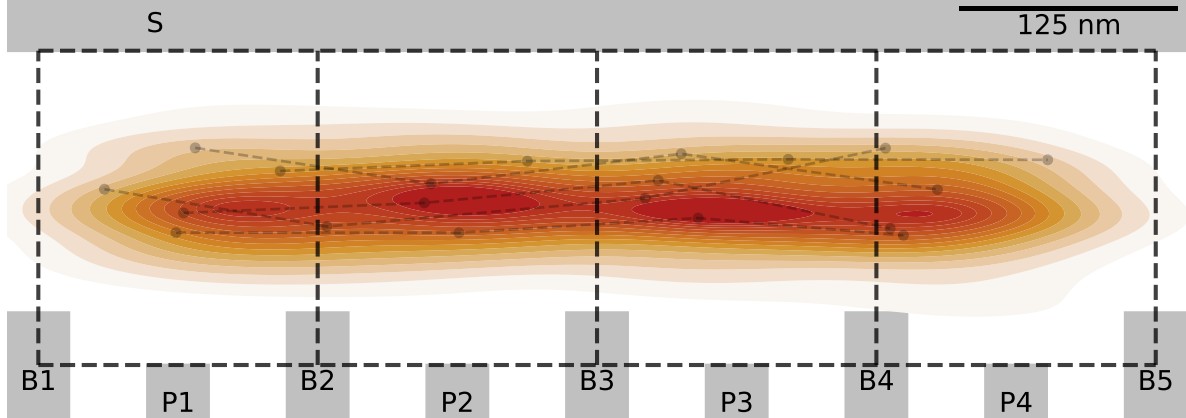

Figure 2: The distribution of dot centers in the samples converged after numerical optimization. Dots connected by dashed lines are the representatives of converged realizations. Each dot is localized to its expected region which is shown by dashed rectangular boxes. Gate profiles are shown similar to Fig. 1.

gate voltages to trap a single electron in each quantum dot and tune $t_{ij}$ to the experimental operating point. The 2DEG electrostatic potential of a tuned clean device is shown in Fig. 1.

Following the analysis of Ref. [32], we focus on the dopant charge distribution as the main source of disorder. We simulate the dopants by making the charge of the mesh volumes in the dopant layer Poisson-distributed random variables with the average charge density equal to the dopant density and consider 200 disorder realizations. The target Hamiltonian for the optimization has $\mu_i = 0\,\text{eV}$ (assuming the Fermi energy in the 2DEG is at $0\,\text{eV}$) and all the interdot couplings are set to $20\,\mu\text{eV}$. The optimization converges to the desired result in 65% of

disorder realizations. While it is likely that further improvements to the optimization algorithm would increase the convergence rate, we consider this performance sufficient since we expect that this value is higher than the experimental yield—the percentage of devices that could be tuned to the proper operating regime compared to the total number of devices attempted in the experiments. The positions of the dot centers in the converged samples exhibit a variation comparable to the dot size, as shown in Fig. 2. We benchmark our numerical simulations by comparing the gate voltages and the gate coupling strengths with the experimentally measured ones in Fig. 3 and Fig. 4 respectively. While the operating point varies strongly across disorder realizations, we find that most measured gate coupling strengths and gate voltages lie within the standard deviation of the computed ones.

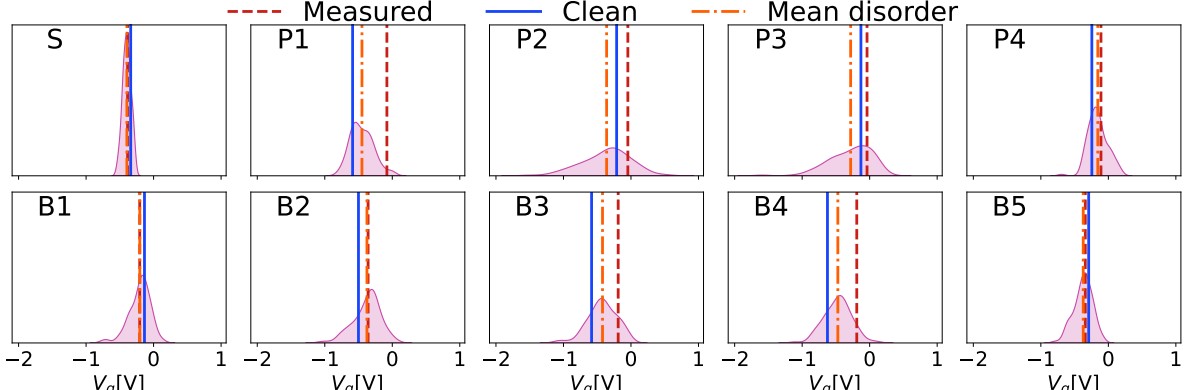

Figure 3: Comparison of gate voltages between the experiment (dashed line) from Ref. [28] and simulation. The distribution of gate voltages for the converged disordered samples is shown. Voltages of the clean system and the mean of the distribution in the disordered case are also shown by solid and dash-dotted lines respectively.

## 3.2 Non-planar geometry

To demonstrate the universality of our approach, we calculate the gate coupling strengths in a non-planar geometry. We consider the Si-heterostructure from Ref. [17, 30]. This device consists of a slab of Si grown on an insulator as shown in Fig. 5. The Si is surrounded by $SiO_2$ on the four faces with plunger and barrier gates deposited on the top, right, and left faces. We tune the gates on one side of the device to form quantum dots underneath. The other side is used for sensing and plays no role in our simulations. Similar to the planar geometry, the plunger gates P1, P2, and P3 control the dot occupation, while the barrier gates B2 and B3 tune the tunnel barrier strengths. We apply the optimization procedure of Sec. 2 to tune the device to its operating point. Due to the device geometry, electron wave functions localize near the sample edge under the plunger gates as shown in Fig. 5.

We simulate disorder in the Si potential landscape by randomly distributing positive charges in $SiO_2$ with a surface charge density of $7.5 \times 10^{10}\,\mathrm{cm}^{-2}$ which is close to the experimentally observed value in the $SiO_2$ [33] and We perform numerical optimization in two stages. Due to the larger effective mass in Si, the tuning of the non-planar device is less stable. We resolve this by excluding the contribution of interdot coupling to $f'(\boldsymbol{V})$ in the first optimization

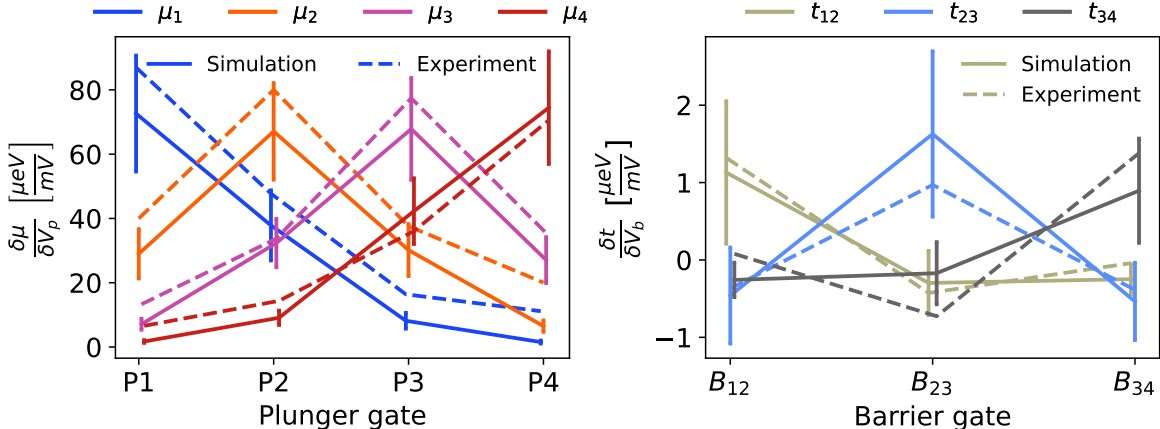

Figure 4: Numerically calculated gate coupling strengths in GaAs heterostructure compared against the measurements from [28]. Solid lines represent the numerical results whereas dashed lines represent the measured data. The mean of the distribution of gate coupling strengths corresponding to the samples converged after numerical optimization is plotted and the error bars indicate the standard deviation. Coupling strengths from plunger gates to the chemical potential of four dots and barrier gates to the interdot tunnel couplings are shown in the top and bottom panels respectively.

step. In the second step, we optimize $f(\boldsymbol{V})$ with $t_0 = 5\,\mu\text{eV}$. We repeat this process for 200 realizations of surface charge disorder. After the two-stage optimization, we compute the gate coupling strengths using Eq. (8) only for those samples for which the optimization converges with precision $|\alpha - 1| < 0.01$, $|\mu_i| < 50\,\mu\text{eV}$ and $|t_{ij} - t_0| < 0.5\,\mu\text{eV}$. Finally, we filter out several outlier disorder realizations with $\delta|t_{ij}|/\delta V > 10\,\mu\text{eV}/\text{mV}$. After performing the two filtering steps, just over 50 % of realizations remain. In Fig. 6, we show the plunger and barrier gate coupling strengths to the dot chemical potential and interdot tunnel couplings for the filtered realizations. We find that due to the stronger localization of the wave functions and smaller distance between the gates and the bound states, the non-local gate couplings are strongly suppressed in the non-planar geometry, compared to the planar one.

## 4   Summary

We developed a software tool to predict the disorder-induced variation in spin qubit devices. Our approach relies on an automated tuning procedure able to tune devices with realistic disorder in the majority of disorder realizations to their operating point. To benchmark our result, we compared the predicted gate voltage and gate coupling strengths distributions to those measured in a GaAs device and found an agreement within a standard deviation of the predicted parameters. We demonstrated that our tool also functions with a comparable efficiency when applied to a non-planar Si device.

    Currently, our package disregards interdot Coulomb repulsion as well as lattice strain. We expect that after extending the simulation to include these phenomena in the simulation, our

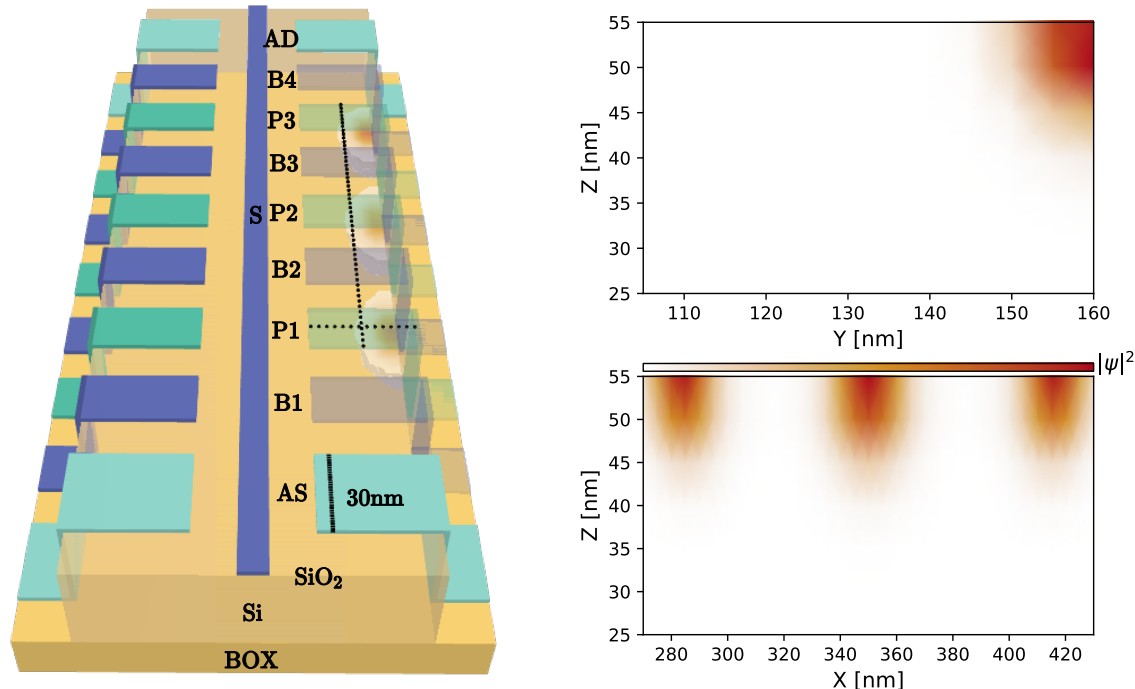

Figure 5: Silicon spin-qubit device with non-planar gate geometry. Si is surrounded on four faces by $SiO_2$ which electrically isolates Si from the gate electrodes. Plungers (P1, P2, and P3) and accumulation (AS and AD) gates are mainly used to accumulate electrons underneath them in Si whereas barrier gates (B0 to B4) are mainly used to form a tunnel barrier between the dots. A similar set of gates shown on the left side of the device is used to form sensor dots. Gate S is used to control the electrostatic coupling between qubits and sensor dots for charge sensing. The probability density of the bound state in three dots on the right channel is shown. Longitudinal cross-section across gates P1 to P3 and lateral cross-section beneath the gate P1 is shown on the right.

approach will become sufficiently general to optimize device geometry and guide the tuning of experimentally realized devices.

# Acknowledgements

We would like to thank I. Araya Day, J. David, C. J. van Diepen, T.-K. Hsiao, A. Lacerda, and S. Miles for useful discussions.

**Data Availability**    All code and data used in the manuscript are available at Zenodo [29].

**Author contributions**    S.R.K. and A.A. formulated the project goal with input from L.W.; A.A. and S.R.K. designed the project approach; S.R.K. and H.K. established electrostatic simulations with input from C.X.L.; under the supervision of A.A., S.R.K. and H.K. carried out numerical simulations of the planar and the non-planar geometry respectively; S.R.K.,

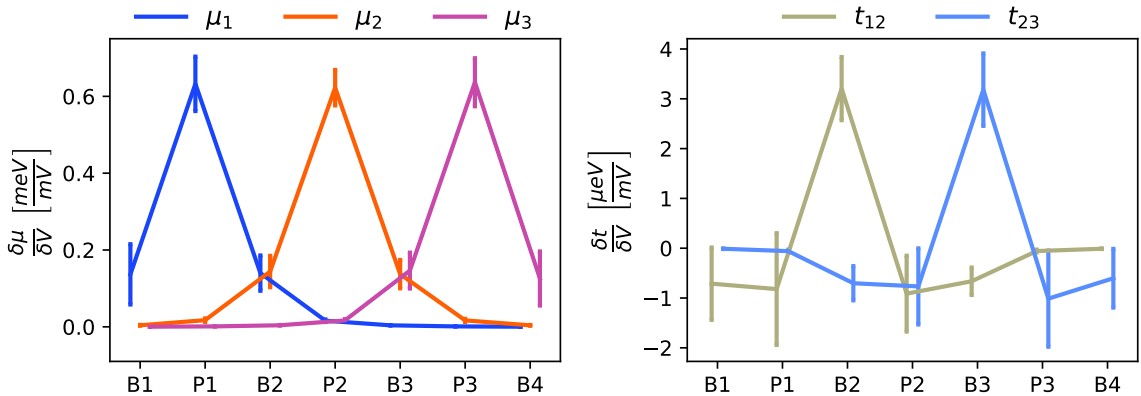

Figure 6: Numerically calculated gate coupling strengths of non-planar geometry. Gate coupling strengths to the chemical potential of three dots and the interdot tunnel couplings are shown in the top and bottom panels respectively. The asymmetric uncertainty in the gate coupling strengths to $\mu_1$ and $\mu_3$ is because the barrier and accumulation gates at the edges of the device are not constrained in the numerical optimization.

C.X.L., and A.A. interpreted the results; S.R.K., H.K., and A.A. wrote the manuscript with input from C.X.L. and L.W.

**Funding information** This work was supported by the the Netherlands Organisation for Scientific Research (NWO/OCW) as part of the Frontiers of Nanoscience program, an ERC Starting Grant 638760, a subsidy for top consortia for knowledge and innovation (TKl toeslag) and a NWO VIDI Grant (016.Vidi.189.180).

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
