# Peer review of "Impact of disorder on the distribution of gate coupling strengths in a spin qubit device"

_SciPost Physics_

## Round 1 · Referee Report · Anonymous (Referee 1) · 2022-12-20

Strengths

1- Concise well-written article. 2- Accompanying code repository.

Weaknesses

1- No new physical result / insight.

Report

The authors develop a spin-qubit device simulation for determining the distribution of the coupling strengths between the electrostatic gate potentials and the effective device Hamiltonian in the presence of disorder. They validate their simulation results with the experimental data of Ref. [28]. Furthermore, they analyze a non-planar geometry inspired by FinFET devices.

One of the problems I see with this manuscript is that the title makes a promise that is not kept. Namely, there is no discussion of the "impact of disorder"; the authors just present simulation results for two experimentally motivated cases, and this is it.

One question that I would have found interesting concerns the kinetic energy in Eq. (1). It seems to me that material properties are taken into account via the effective mass $m_e$, but the band structure of the compound is otherwise ignored, although it could be taken into account via the tight-binding model that the authors actually investigate.

I have some further relatively minor comments that I list under "Requested changes".

My overall impression is that this submission is unsuitable for SciPost Physics. However, with some minor changes, it might fit into SciPost Physics Core. Alternatively, if the authors really do care about their simulations code, they might consider transferring to SciPost Physics Codebases.

Requested changes

1- Reconsider the title. 2- Discuss approximations made to the band structure by the Hamiltonian Eq. (1). 3- There are two instances of $H$ where in my opinion math mode seems to be missing (2 lines before Eq. (5) and line after Eq. (6)), same for $x$ (line before Eq. (5)) and $k$ (line before Eq. (7)). 4- 3rd line of second paragraph of section 3.2: There is an upper case "We" in the middle of a sentence (lowercase?). 5- Left panel of Fig. 5: the "S" is difficult to see. Indeed, black text on blue background has bad contrast and would thus better be avoided. 6- The caption of Fig. 5 refers to a "B0" that I was unable to find. 7- Duplicate "the" on the first line of the "Funding information". 8- There are duplicates of DOI links in some references ([5,14,15,20,23,26,32]). 9- Ref. [18] is published in Nature Electronics 5, 184 (2022).

---

## Editorial Decision

awaiting_resubmission